# Thermal Calibration of Triaxial Accelerometer for Tilt Measurement

**DOI:** 10.3390/s23042105

**Published:** 2023-02-13

**Authors:** Bo Yuan, Zhifeng Tang, Pengfei Zhang, Fuzai Lv

**Affiliations:** 1Polytechnic Institute, Zhejiang University, Hangzhou 310027, China; 2College of Biomedical Engineering and Instrument Science, Zhejiang University, Hangzhou 310027, China; 3State Key Laboratory of Fluid Power and Mechatronic Systems, School of Mechanical Engineering, Zhejiang University, 38 Zheda Road, Hangzhou 310027, China

**Keywords:** MEMS, triaxial accelerometer, error analysis, thermal drift, calibration, temperature compensation, tilt measurement

## Abstract

The application of MEMS accelerometers used to measure inclination is constrained by their temperature dependence, and each accelerometer needs to be calibrated individually to increase stability and accuracy. This paper presents a calibration and thermal compensation method for triaxial accelerometers that aims to minimize cost and processing time while maintaining high accuracy. First, the number of positions to perform the calibration procedure is optimized based on the Levenberg-Marquardt algorithm, and then, based on this optimized calibration number, thermal compensation is performed based on the least squares method, which is necessary for environments with large temperature variations, since calibration parameters change at different temperatures. The calibration procedures and algorithms were experimentally validated on marketed accelerometers. Based on the optimized calibration method, the calibrated results achieved nearly 100 times improvement. Thermal drift calibration experiments on the triaxial accelerometer show that the thermal compensation scheme in this paper can effectively reduce drift in the temperature range of −40 °C to 60 °C. The temperature drifts of x- and y-axes are reduced from −13.2 and 11.8 mg to −0.9 and −1.1 mg, respectively. The z-axis temperature drift is reduced from −17.9 to 1.8 mg. We have conducted various experiments on the proposed calibration method and demonstrated its capacity to calibrate the sensor frame error model (SFEM) parameters. This research proposes a new low-cost and efficient strategy for increasing the practical applicability of triaxial accelerometers.

## 1. Introduction

Micro-electromechanical systems (MEMS) have been evolving since MEMS-based inertial sensors were widely used in commercial and military applications such as human motion tracking, navigation systems, and posture control systems [1,2,3,4]. Features such as tiny size, cost, and low energy consumption make them very attractive. Such accelerometers must be calibrated as precisely as possible because even slight biases in accelerometers can result in inaccurate position measurement when used for inertial navigation system (INS) applications and inaccurate tilt angle measurements [5,6,7]. Furthermore, the performance of these sensors is greatly dependent on external circumstances, such as temperature changes. As a result, precise, reliable, and effective thermal models are required to decrease the impact of these errors, which can degrade system performance [8,9,10,11]. Factory-based triaxial accelerometers calibration is a time-consuming and expensive process, usually performed for specific advanced sensors [12].

There are several different errors and temperature-dependent models for triaxial MEMS sensors and corresponding calibration methods based on different principles, but they have restrictions, for instance, the need to provide accurate platforms for precise alignment or other temperature compensation devices. It is not sufficient for manufacturers to perform only basic calibration of low-cost sensors, as even small uncompensated coefficients can lead to inaccurate tilt angle estimation and increased bias in position measurements [13,14,15,16]. These requirements increase manufacturing costs. Therefore, it is necessary to investigate alternative methods [17].

For example, the common accelerometer calibration method described by Khan and Ranj [18] uses six specific positions where the sensor axes are precisely aligned along the axis of the calibration device. The accelerometer is calibrated by a specific position with a certain reference angle. This calibration method can only estimate offset and scale factor errors, not non-orthogonality, and the accuracy of the calibration depends heavily on the accuracy of alignment with the calibration device [19]. The ellipsoidal fitting method is often used to improve calibration accuracy while reducing device costs, as presented in [20,21]. This method combines the error parameters by fitting them to the ellipsoid data through linear regression, and then based on the calculated ellipsoid parameters, the unknown error model parameters can be estimated. Maximum likelihood estimation tackles the problem from a probabilistic perspective, but usually uses probability density functions rather than probabilities [22,23]. The Kalman filter and its extensions attempt to estimate both the error model and the calibration parameters within a calibration function based on the system model and observations over a period of time [24].

In terms of temperature calibration, the manufacturer’s technical data are not sufficient for inclination measurement applications. The target of temperature calibration is to determine sensor errors at various temperature conditions. There are two prevailing methods regarding temperature calibration: the soak method and the ramp method. The soak method assumes a stable sensor temperature, while the ramp method assumes the sensor temperature varies with time [25,26,27]. Talha, Kivanc and Tayfun described a temperature compensation method for a condenser MEMS accelerometer by using a MEMS dual-ended tuning fork to compensate for the temperature dependence of the accelerometer’s output [28]. The purpose of temperature compensation is to determine the error of the triaxial accelerometers at different temperature points. Xu described an efficient temperature compensation methodology based on neural networks using the collected accelerometer responses from −30 °C to 50 °C as learning data [29]. In [30], Guo proposed a temperature calibration method based on reservoir computing with a MEMS resonator. The novel real-time online temperature compensation method can achieve high prediction accuracy.

These methods can be divided into the 3-order polynomial fitting method [31], the linear interpolation method [32], the AG-based calibration method [33], the RBF calibration method [34], and other optimization methods. In this research, we apply the soak method for triaxial accelerometers based on the Levenberg-Marquardt (LM) algorithm and polynomial methods, which can provide more accurate sensor error data at various temperature points. Furthermore, several solutions to its inherent issues are considered.

This paper is organized as follows. In Section 2, we introduce the error source of the triaxial accelerometer and describe the error model of the three-axis MEMS accelerometer, where the biases, scale factors, nonorthogonal error, and misalignments are considered thermal calibration parameters. We describe the algorithms used for its calibration and temperature compensation in Section 3. Experiments and analysis are provided in Section 4.

## 2. Error Source and Error Model

An accelerometer is a device that can detect acceleration in accordance with Newton’s second law of motion [35]. It is made up of a seismic mass as well as a capacitance to voltage converting circuit. When the acceleration acts on the accelerometer, the seismic mass deflects, causing a change in capacitance as well as voltage. As with other sensors, triaxial accelerometers are subject to measurement errors due to variations in internal structure and external environment. The errors of triaxial accelerometers can be roughly divided into deterministic errors and random errors. Deterministic errors include zero bias errors, scale factor, and non-orthogonality errors. Random errors are caused by drift errors, random noise, and turn-on errors, which cannot be predicted or compensated directly [36,37,38].

### 2.1. Sensor Frame Error Model

As shown in Figure 1a, the output signal from the MEMS accelerometers should be zero when there is no signal input to the sensor, but this is not the case. This phenomenon is caused by a flaw in the manufacturing of the folded cantilever that holds the accelerometer’s mass. The asymmetrically folded cantilever fails to bring the mass into equilibrium. As a result, a false differential capacitance influences the sensor electrode. A faulty suppression of parasitic capacitance in a capacitive sensor, on the other hand, introduces uncompensated biases. As shown in Figure 1b, the scale factor is the ratio of an output change to an intended input variation. An ideal sensor has only one scale factor. The scale factor error can be introduced by flaws in the manufacturing of the folded cantilever and the process of weak signal detection and extraction. Each sensitive axis’ scale factor can be expressed as a series of coefficients. As shown in Figure 1c, ideally, the triaxial accelerometer sensitivity axes should be orthogonal, but inaccuracies in the internal construction of the chip can cause non-orthogonal errors between the axes. MEMS sensors output a voltage that is proportional to the acceleration detected by the sensor. As shown in Figure 1d, when the triaxial accelerometer is installed in an inclinometer, the three sensitive axes should be aligned with the three orthogonal axes of the body frame. In practice, however, misalignment errors occur due to errors in the sensor mounting process [39,40]. To ensure that the orthogonal output readings are displayed correctly with the sensor frame by the nonorthogonal triaxial accelerometer, misalignment errors *θ*, *φ*, and *ψ* must be estimated.

For triaxial accelerometers, we define the SFEM for the calibration of triaxial MEMS accelerometers. Based on the above analysis of various error characteristics of the triaxial accelerometers, the SEFM can be defined as:(1)$$\begin{array}{c} U_{\mathrm{SF}}=M_{\mathrm{SF}}C_{\mathrm{SF}}S_{\mathrm{SF}}\;(U-b_{\mathrm{SF}})\\ =\left(\begin{array}{ccc} m_{\mathrm{xx}} & m_{\mathrm{xy}} & m_{\mathrm{xz}}\\ m_{\mathrm{yx}} & m_{\mathrm{yy}} & m_{\mathrm{yz}}\\ m_{\mathrm{zx}} & m_{yz} & m_{\mathrm{zz}} \end{array}\right)\left(\begin{array}{ccc} 1 & 0 & 0\\ \alpha_{\mathrm{yx}} & 1 & 0\\ \alpha_{\mathrm{zx}} & \alpha_{\mathrm{zy}} & 1 \end{array}\right)\left(\begin{array}{ccc} S_{\mathrm{ax}} & 0 & 0\\ 0 & S_{\mathrm{ay}} & 0\\ 0 & 0 & S_{\mathrm{az}} \end{array}\right)\left(\left(\begin{array}{c} U_{x}\\ U_{y}\\ U_{z} \end{array}\right)-\left(\begin{array}{c} b_{x}\\ b_{y}\\ b_{z} \end{array}\right)\right) \end{array}$$
where U_SF_ = ${\left[X_{\mathrm{SF}},Y_{\mathrm{SF}},Z_{\mathrm{SF}}\right]}^{T}$ is the vector of accelerations after calibration; M_SF_ is the matrix describing the misaligned errors; C_SF_ is the matrix providing the conversion from non-orthogonal to orthogonal frames with non-diagonal angles $\alpha_{\mathrm{yx}}$, $\alpha_{\mathrm{zx}}$, and $\alpha_{\mathrm{zy}}$ corresponding to the triaxial misalignment; S_SF_ is a scale factor parameter matrix; U =$\;{\left[U_{x},U_{y},U_{z}\right]}^{T}$ is the vector of raw sensor readings; b_SF_ $={\left[b_{x},b_{y},b_{z}\right]}^{T}\;$ is the vector of sensor offsets.

### 2.2. Temperature Dependence

Thermal drift of MEMS errors are typically considered deterministic errors. The following two methods are required to decrease the thermal drift of sensor errors:(1)Temperature calibration: developing accurate and reliable thermal models of the sensor errors, i.e., establishing a relationship between the sensor errors and the sensor temperature;(2)Temperature compensation: compensating the thermal drift of the sensor errors based on their temperature during the accelerometer’s operation process. Both of these processes are dependent on the temperature generated by the accelerometer’s internal temperature sensors.

Considering thermal errors, because temperature variations affect the internal structure of the triaxial accelerometer, the error parameters of the triaxial accelerometer vary nonlinearly with temperature. Considering the temperature factor, we redefine Equation (1) as
U_SF_(T) = M_SF_(T)C_SF_(T)S_SF_(T) (U − b_SF_(T))(2)

In the above formula, T represents the temperature measured by the thermal sensor. M_SF_(T), S_SF_(T), C_SF_(T), and b_SF_(T) are the coefficients that change with temperature in Equation (1). U_SF_(T) represents the output data at temperature T, and U is the vector of raw sensor readings.

## 3. Calibration Method

This section relies on triaxial accelerometer calibration and temperature compensation algorithms. The basic principle of calibration and compensation is based on the principle that triaxial acceleration should be calculated on the same order of magnitude as gravity in Equation (3):(3)$$G=\sqrt{X_{\mathrm{SF}}^{2}{+Y}_{\mathrm{SF}}^{2}{+Z}_{\mathrm{SF}}^{2}}$$
where $X_{\mathrm{SF}}^{2},Y_{\mathrm{SF}}^{2},\;$and $Z_{\mathrm{SF}}^{2}$ are the accelerations present in the sensor frame axes; G is the gravitational acceleration, ideally equal to 1 g.

### 3.1. Levenberg-Marquardt Algorithm

The Levenberg-Marquardt (LM) algorithm is one of the most efficient and widely used algorithms and is often used to solve nonlinear least squares problems. It is more robust than the Gauss-Newton (GN) algorithm. The Levenberg-Marquardt algorithm combines two numerical optimization algorithms: the Gradient Descent (GD) method and the Gauss-Newton (GN) method [41,42,43]. To obtain the most precise coefficients without using a highly accurate turntable system, the triaxial accelerometer should be placed continuously and fixed to cover the entire surface of the sphere, and the accelerometer should be influenced only by gravity during the experiment. However, this is not possible in practice, because the measurement sum for this calibration method would be infinite. So, we tried to test and optimize the number of positions for compensation during the calibration. In [44], the 24-position method was used, as shown in Table 1 for three directions along the x, y, and z axes, with eight positions in each direction. The calibration method can be described as (4):(4)$$S\left(\beta \right)=\overset{m}{\underset{i=1}{\sum }}[y_{i}-f(x_{i},\beta ){]}^{2}$$
where S(β) denotes the sum of $[y_{i}-f\left(x_{i},\beta \right){]}^{2}$; m is the measurement number; $x_{i}$ is the measured value; $y_{i}$ are the reference acceleration data, and β is a parameter vector as defined in Equation (1).

Here we use the Levenberg-Marquardt algorithm, which reduces S(β) with respect to the parameters in vector β.
(1)The Gradient Descent Method: The Gradient Descent (GD) algorithm is a commonly used minimization method that updates the parameter values in the opposite direction of the gradient from the objective function. The GD algorithm is highly convergent and can be used for optimization problems with thousands of parameters. The GD algorithm update $h_{\mathrm{gd}}$ that modifies the S(β) in the direction of the steepest descent can be defined as shown in Equation (5).
(5)$$h_{\mathrm{gd}}={\alpha J}^{T}W\left(y_{i}-f\left(x_{i},\beta \right)\right)$$
where α is a positive scalar corresponding to the step in the steepest-descent direction; J is a Jacobian matrix based on the vector β; W can be set as the inverse matrix of the measurement error covariance matrix.(2)The Gauss-Newton Method: The Gauss-Newton method is a method for the minimization of the sum-of-squares target function. For medium-sized problems, the Gauss-Newton method usually converges faster than the gradient descent method. The formula for the GN algorithm to reduce S(β) is given by the following Equation (6).
(6)$$[J^{T}\mathrm{WJ}]\;h_{\mathrm{gn}}=J^{T}W\left[y_{i}-f\left(x_{i},\beta \right)\right]$$
where $h_{\mathrm{gn}}\;$ indicates the GN algorithm update of the parameter estimated to lead to the minimization of S(β).(3)The Levenberg-Marquardt method: The Levenberg-Marquardt algorithm adaptively changes the parameter updates between gradient descent parameter iterations and Gauss-Newton parameter iterations to achieve optimal progress in the minimization of S(β). The LM algorithm can be described by Equation (7).
(7)$$\left[J^{T}\mathrm{WJ}+\lambda \mathrm{diag}\left(J^{T}\mathrm{WJ}\right)\right]h_{\mathrm{lm}}=J^{T}W\left[y_{i}-f\left(x_{i},\beta \right)\right]$$
where J is the Jacobian matrix of vectors β; W is the weighting diagonal matrix; $h_{\mathrm{lm}}\;$is adaptively weighted to reach optimal progress in S(β) minimization. The damping factor λ is adjusted at each iteration.


### 3.2. Least-Squares Fitting of Data by Polynomials

The temperature dependence of MEMS accelerometers can be defined as the non-linear change of calibration parameters with temperature [45]. For temperature compensation, we convert the error coefficients into variables that change with temperature, which will vary according to the environment and other instrumental conditions. Polynomial regression methods are utilized in this research to model triaxial accelerometer errors based on temperature [46]. The calibration parameters in Equation (1) can be fitted with a three-dimensional polynomial as shown in Equation (8).
(8)$$p\left(t\right)=a_{0}+a_{1}f\left(t\right)+a_{2}f\left(t^{2}\right)+a_{3}f\left(t^{3}\right)$$
where p(t) is the calibration coefficients in (1); t is the temperature coefficient; $a_{0\;},a_{1\;},a_{2\;},a_{3}$ are the fitting coefficients. The temperature calibration procedure can be found in Figure 2.

## 4. Experiments and Results

### 4.1. Calibrated Sensors and Measurement Setup

In this section, we will briefly describe the system (Figure 3) used for calibration and temperature compensation. The performance parameters of the triaxial accelerometer are shown in Table 2. A three-axis position turntable equipped with a thermal chamber (see Figure 4 is used to calibrate the triaxial accelerometer sensor errors over a wide temperature range. The triaxial accelerometer is fixed in the center of the triaxial turntable by using a clamp, and the turntable is controlled according to the programmed control commands. The device specifications for the turntable and the thermal chamber are shown in Table 3.

### 4.2. Sensor Frame Error Model Analyses

For compensation purposes and sequential testing, we measured the raw data of the accelerometers at 1296 locations distributed uniformly over the sphere (Figure 5). This number is based on the number of calibration positions suggested in Section 3 multiplied by 54 for calibration position optimization. Then we analyze the dependence of the calibrated SFEM on the number of calibration positions.

The three-axis sensor is oriented in each directi on as measurements are taken. The calibration device is set to stay at each position for a while and calculate the average of 200 measurement data samples to minimize sensor noise [47]. The impact of calibration on the measured data is shown in Figure 6 and Figure 7, where the deviation of the calibrated acceleration reading from 1g is approximately 100 times smaller than the data before calibration.

Calibration parameters are estimated and used to remove the errors from the raw data. Figure 7 displays the raw data as well as the final calibrated data, together with the estimation of the ellipsoid and sphere [48].

The LM algorithm was used to estimate the SFEM from the measurement values of the proposed positions. It contributed to reducing the influence of manufacturing defects on sensor accuracy. To achieve the optimization of the number of positions, the Root Mean Square Error (RMSE) of the comparison defined in (9) was used:$$\mathrm{RMSE}=\sqrt{\frac{\sum_{i=1}^{n}{\left(x_{i}-g\right)}^{2}}{n}}$$
(9)$$x_{i}=\sqrt{g_{xi}^{2}+g_{yi}^{2}+g_{zi}^{2}}$$
where n is the number of positions; g is numerically the same as gravity and is equal to 1 g; $g_{xi}$, $g_{yi}$, and $g_{zi}\;$are the components of gravity along each axis.

To demonstrate that 24 positions are adequate for compensation goals, we evaluated the RMSE of the SFEM from 12 to 1296 for various numbers of positions (NoP). The results are shown in Figure 8. In Table 4, the NoP can be found, where N indicates the quantitative relationship between the axis and the NoP used for the analysis [49].

As can be seen in Figure 8, 21 or more positions have the capacity to meet the required requirements, regardless of how many positions are rotated on the calibration device. This can also be interpreted as a variation of the compensation results below the controllable value when 21 or more positions (up to 1296) are used; therefore, we can optimize the number of calibration positions by using 24 positions to cover all axes, which is the number we used in Table 1.

After calibrating our triaxial accelerometer sensor, we fused the sensor measurements for attitude estimation using the method described in [46]. To further verify the effectiveness of the calibration, another analysis was performed where the tilt angle estimated from the calibration was compared to the reference angle measured by the Rotating Tilt Device (RTD). As shown in Figure 4b, we mounted the accelerometer on the RTD and tilted it by a specific angle along both axes. Tilt sensing measures the angle of change with respect to gravity. The output of the accelerometer sensor is read by the microcontroller’s internal Analog to Digital Converter (ADC) to determine tilt. The tilt of each axis can then be calculated using the following formula (10). The tilt corresponds to the pitch angle and roll angle. The specifications are listed in Section 4 for the RTD.
$$\varphi_{m}=\arctan\left(-Y_{\mathrm{SF}}/\sqrt{X_{\mathrm{SF}}^{2}+Z_{\mathrm{SF}}^{2}}\right)$$
(10)$$\theta_{m}=\arctan\left(X_{\mathrm{SF}}/-Z_{\mathrm{SF}}\right)$$
where $\varphi_{m}$ is the pitch angle; $\theta_{m}$ is the roll angle; $X_{\mathrm{SF}},Y_{\mathrm{SF}},Z_{\mathrm{SF}}$ are the measurement of the acceleration. The comparison of the tilt angle measurements before and after calibration can be seen in Table 5, where the tilt angle is more accurate than the angle without calibration due to calibration. In the last column, we use the Sensor Error Optimization Quantity (SEOQ), which is defined as the specific deviation for the difference between the two values with respect to the maximum angle, i.e., 30 degrees.

### 4.3. Temperature Compensation

Based on the 24-position data measurement method described in Section 3 and demonstrated in Section 4, we conducted experiments (24 positions in our case) with temperatures cycling from −40 °C to 60 °C at 10 °C intervals, verifying the calibration coefficients at each temperature point (Figure 9).

The triaxial accelerometer is enclosed in a thermal chamber and given enough time (approximately one hour) to stabilize its temperature at the set temperature point in the temperature compensation method. The system begins recording data once the accelerometer and thermal chamber temperature have stabilized [50]. Figure 10 shows the relationship between temperature and calibration. Due to the temperature hysteresis characteristics, we fit scatter points for temperature compensation. The analyses of the other coefficients were very similar.

To verify the calibration results at specific temperatures, we performed the 24-position method at some temperatures and compared the RMSE before and after calibration, as shown in Table 6.

To verify the effect of the temperature compensation in a dynamic temperature environment, we fixed the accelerometer in a temperature-controlled oven, heated it to 60 °C, and then closed the temperature oven to allow the accelerometer to cool down naturally [51]. The compensated and uncompensated accelerometer output data can be examined in Figure 11. The temperature dependency is reduced from 1346 µg/K to 37 µg/K.

## 5. Conclusions

The main motivation of this paper is to analyze the validity of calibration and temperature compensation methods. We evaluated the SFEM for the ADXL355 accelerometer based on the Levenberg-Marquardt (LM) algorithm and polynomial methods. Ideally, the value of the triaxial accelerometer should be equal to the gravitational force in the static case, whether the temperature conditions are considered or not. The LM algorithm proposed in this paper is robust and efficient for the calibration of triaxial accelerometers. To compensate for thermal changes, the temperature compensation method was used combined with real-time temperature monitoring to model and incorporate temperature-related drift characteristics. Various experiments were performed to demonstrate different aspects of calibration and temperature compensation, for example, how many positions must be used to achieve the accuracy we require and how to prove the effectiveness of temperature compensation. The experimental results show that this calibration method is effective in reducing positioning calculation errors. In all cases, there is an improvement of about 100 times after calibration, and the variation of the error parameters is reduced by almost 80% after temperature compensation. All results demonstrate the applicability of the proposed calibration and temperature compensation methods.

Some improvements can be made to the model of the triaxial accelerometer. When using the thermal calibration method to calculate the bias, scale factor, and misalignment errors at each temperature point, the results may be inaccurate since the sensor takes a long time to stabilize at the temperature being measured. More circumstances, such as input accelerations and temperature change rates that fluctuate over time, should be considered in the future. Therefore, multiple thermal models calibrated in various ambient temperatures should be developed, and a combined thermal model could be employed for demanding applications.

## Figures and Tables

**Figure 1 sensors-23-02105-f001:**
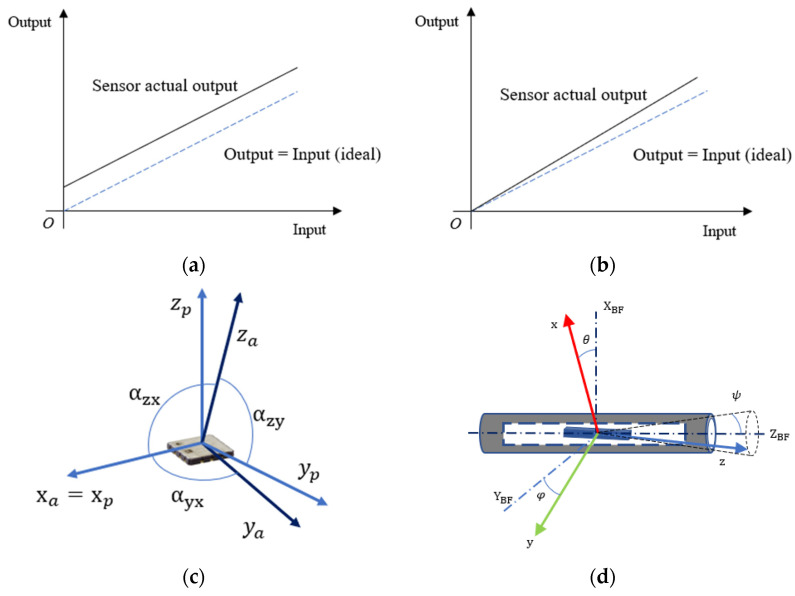
Sensor frame error model of triaxial accelerometer. (**a**) Zero bias error; (**b**) scale factor error; (**c**) nonorthogonality error; (**d**) misalignment error.

**Figure 2 sensors-23-02105-f002:**
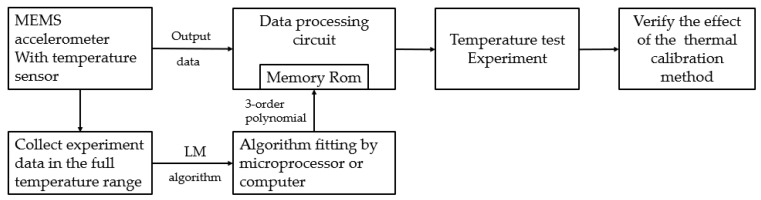
The procedure of temperature calibration.

**Figure 3 sensors-23-02105-f003:**
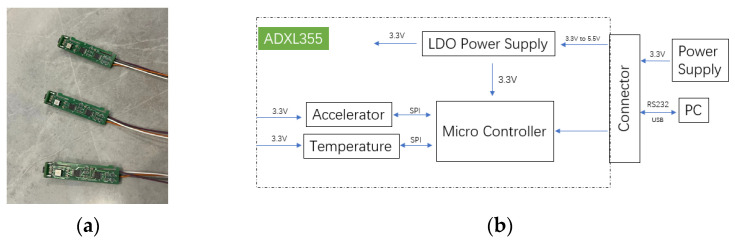
Measurement setup for triaxial accelerometer calibration. (**a**) ADXL355 sensors; (**b**) calibration and measurement device for triaxial accelerometer.

**Figure 4 sensors-23-02105-f004:**
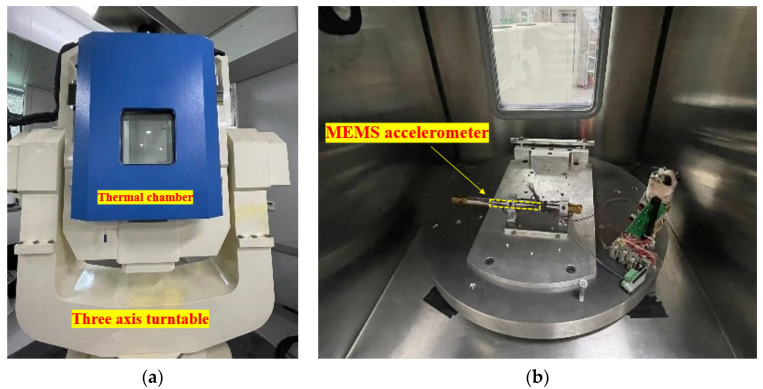
Temperature-controlled rotational-tilt platform. (**a**) External view; (**b**) internal view.

**Figure 5 sensors-23-02105-f005:**
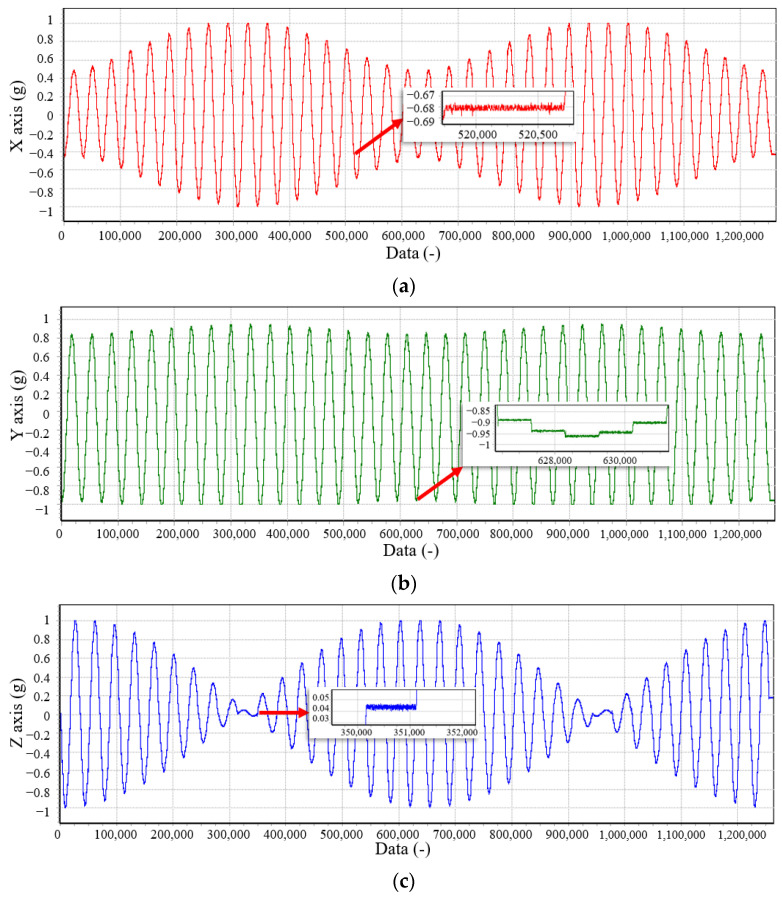
Acceleration values during the proposed 1296 positions calibration method. (**a**) Raw data of the x-axis; (**b**) raw data of the y-axis; (**c**) raw data of the z-axis.

**Figure 6 sensors-23-02105-f006:**
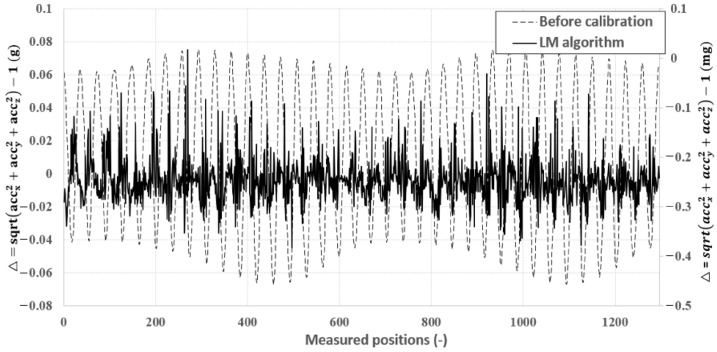
Analysis of the calibration between acceleration deviations observed at various points before (left vertical axis) and after (right vertical axis) calibration.

**Figure 7 sensors-23-02105-f007:**
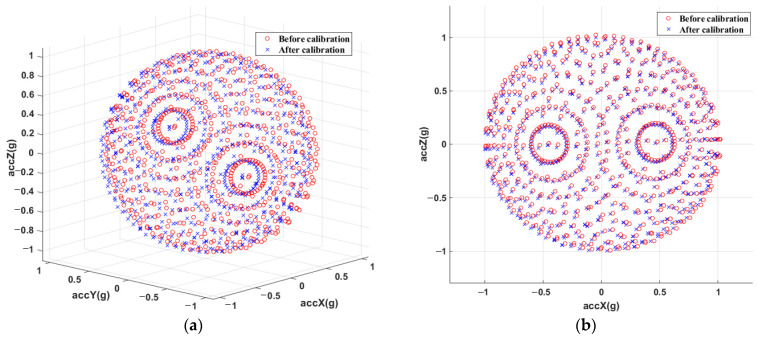
The 3D graphics of measured accelerations before (red ‘o’) and after (blue ‘x’) calibration. (**a**) main view; (**b**) front view; (**c**) end view; (**d**) vertical view.

**Figure 8 sensors-23-02105-f008:**
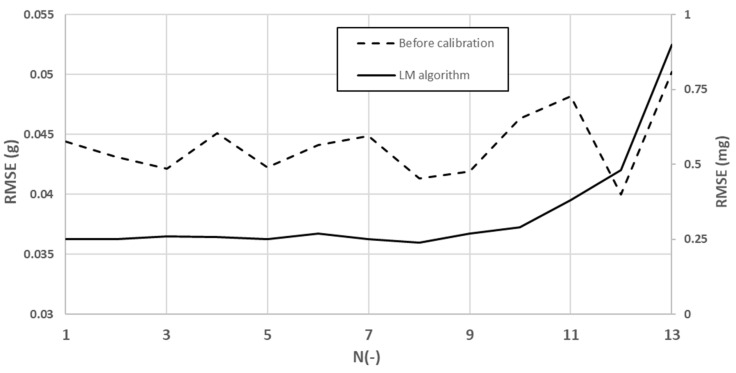
Relationship of RMSE before calibration (left vertical axis) and after calibration (right vertical axis).

**Figure 9 sensors-23-02105-f009:**
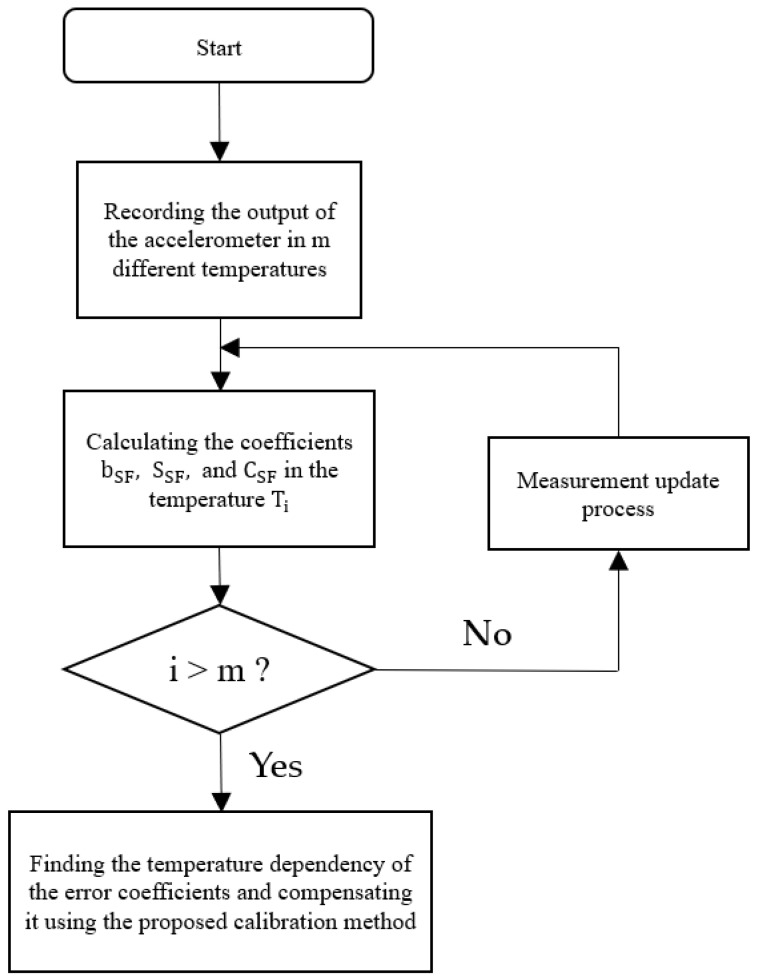
The estimation program routine of the triaxial accelerometer calibration method.

**Figure 10 sensors-23-02105-f010:**
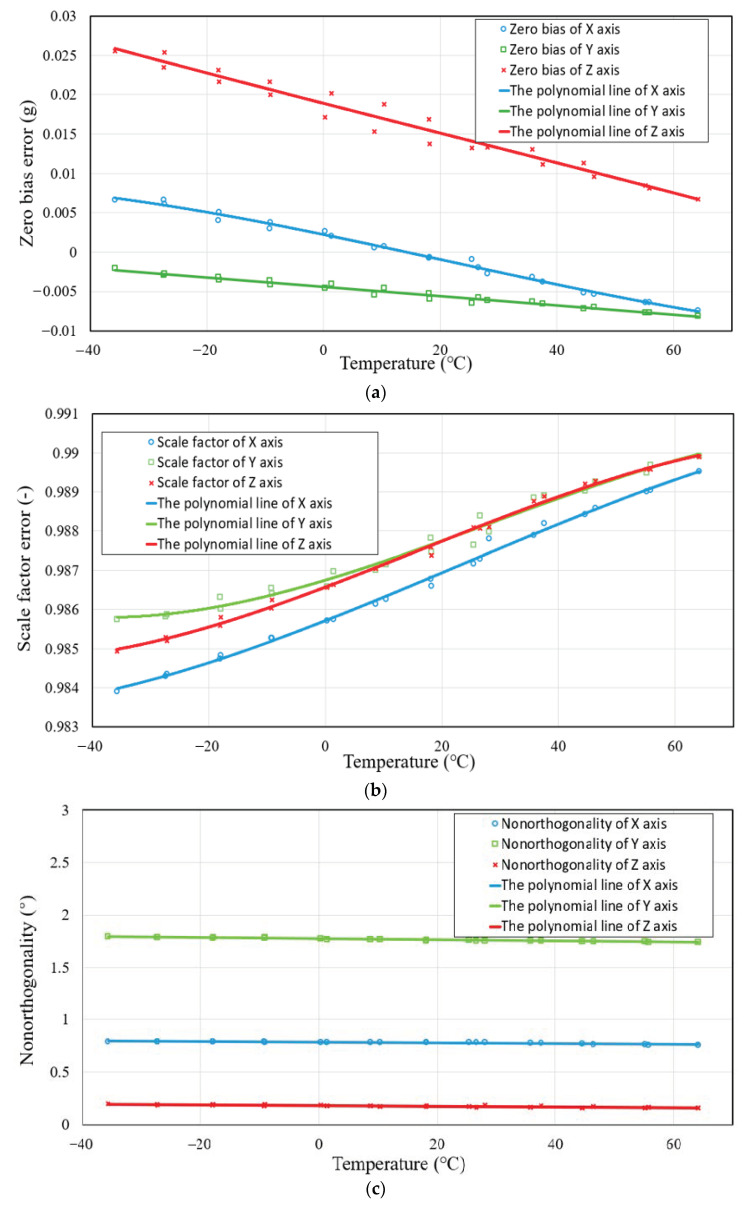
The polynomial line of each sensor axis. (**a**) Zero bias error; (**b**) scale factor error; (**c**) nonorthogonality error.

**Figure 11 sensors-23-02105-f011:**
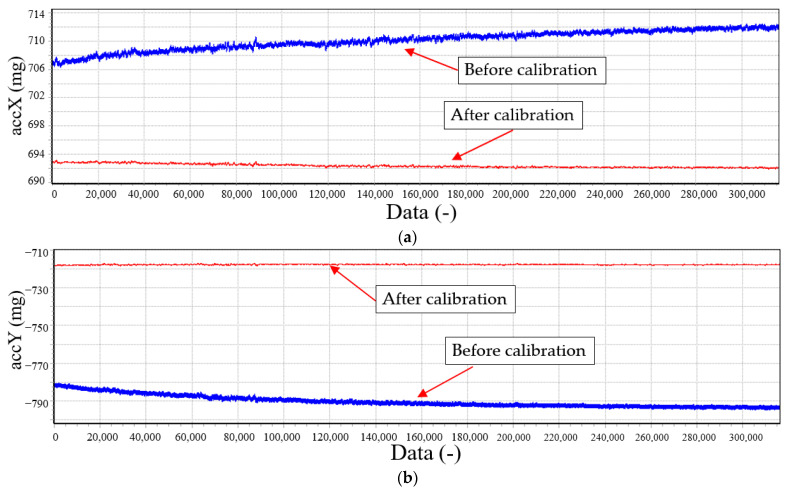
Dependence of temperature before (blue line) and after (red line) calibration. (**a**) Thermal calibration of the x-axis. (**b**) Thermal calibration of the y-axis. (**c**) Thermal calibration of the z-axis.

**Table 1 sensors-23-02105-t001:** Suggested Calibration Positions.

No.	X Axis Position	Y Axis Position	Z Axis Position
1	P_x1_ ≈ horizontal	P_y1_ ≈ vertical	P_z1_ ≈ horizontal
2	P_x1_ + 45°	P_y1_ + 45°	P_z1_
3	P_x1_ + 90°	P_y1_ + 90°	P_z1_
4	P_x1_ + 135°	P_y1_ + 135°	P_z1_
5	P_x1_ + 180°	P_y1_ + 180°	P_z1_
6	P_x1_ + 225°	P_y1_ + 225°	P_z1_
7	P_x1_ + 270°	P_y1_ + 270°	P_z1_
8	P_x1_ + 315°	P_y1_ + 315°	P_z1_
9	P_x2_ ≈ horizontal	P_y2_ ≈ horizontal	P_z2_ ≈ vertical
10	P_x2_ + 45°	P_y2_	P_z2_ + 45°
11	P_x2_ + 90°	P_y2_	P_z2_ + 90°
12	P_x2_ + 135°	P_y2_	P_z2_ + 135°
13	P_x2_ + 180°	P_y2_	P_z2_ + 180°
14	P_x2_ + 225°	P_y2_	P_z2_ + 225°
15	P_x2_ + 270°	P_y2_	P_z2_ + 270°
16	P_x2_ + 315°	P_y2_	P_z2_ + 315°
17	P_x3_ ≈ horizontal	P_y3_ ≈ horizontal	P_z3_ ≈ vertical
18	P_x3_	P_y3_ + 45°	P_z3_ + 45°
19	P_x3_	P_y3_ + 90°	P_z3_ + 90°
20	P_x3_	P_y3_ + 135°	P_z3_ + 135°
21	P_x3_	P_y3_ + 180°	P_z3_ + 180°
22	P_x3_	P_y3_ + 225°	P_z3_ + 225°
23	P_x3_	P_y3_ + 270°	P_z3_ + 270°
24	P_x3_	P_y3_ + 315°	P_z3_ + 315°

**Table 2 sensors-23-02105-t002:** Characteristics of tested ADXL355.

Specification	Value
Interface	Digital
Noise-Density (μg/$\sqrt{\mathrm{Hz}}$)	25
0 g Offset (mg)	±25
Range (g)	±2/±4/±8
ADC	20-bit
Output Data Rate (Hz)	0~4000 Hz

**Table 3 sensors-23-02105-t003:** Performance of thermal calibration equipment.

Turntable	
Principal axis rotation range	Continuous infinite
Tilting axis angular position accuracy	±3″
Principal axis angular position accuracy	±3″
Non-orthogonalities between axes	±5″
Tilting axis rotation range	±90°
**Thermal Chamber**	
Temperature range	−55~+100 °C
Temperature change rate	±0.1~±5 °C/min linear

**Table 4 sensors-23-02105-t004:** Relationship between the NoP and the axes.

N	NoP	N	NoP	N	NoP
1	1296	6	64	11	18
2	648	7	48	12	16
3	324	8	36	13	12
4	162	9	24		
5	81	10	21		

**Table 5 sensors-23-02105-t005:** Comparison of tilt angles before and after calibration.

Reference Angle φ; θ (deg)	Without Calibration φ; θ (deg)	After Calibration φ; θ (deg)	SEOQ φ; θ (%)
0; 0	−0.77; 0.59	−0.70; −0.30	0.2; 3.0
15; 0	14.18; −0.62	14.61; −0.15	1.4; 1.6
30; 0	29.18; −0.63	30.13; 0.01	3.2; 2.1
0; −15	−0.96; −15.83	−0.91; −15.72	0.2; 0.4
0; −30	−0.82; −31.10	−0.76; −31.00	0.2; 0.3
15; −15	14.31; −15.98	14.76; −15.82	1.5; 0.5
30; −30	29.00; −29.62	29.98; −29.76	3.3; 0.5

**Table 6 sensors-23-02105-t006:** Comparison of RMSE before and after calibration.

Temperature (°C)	Before Calibration RMSE (mg)	Polynomial RMSE (mg)	Measurement RMSE (mg)
−35	45.1	0.19	0.17
−25	46.8	0.24	0.21
−15	47.2	0.22	0.19
−5	45.9	0.24	0.21
5	44.3	0.18	0.17
15	46.3	0.19	0.21
25	48.6	0.25	0.25
35	47.2	0.26	0.29
45	45.3	0.29	0.24
55	48.1	0.31	0.29

## Data Availability

The original signals and test signals presented in this study are available on request from the corresponding author.

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
