# Peer review of "Thermal Calibration of Triaxial Accelerometer for Tilt Measurement"

_sensors, 2023, doi:10.3390/s23042105_

Round 1

Reviewer 1 Report

This manuscript presents an algorithm-based method for the calibration of triaxial accelerometer with thermal compensation. By using 24-position data measurement and temperature compensation, about 100 times calibration improvement is demonstrated in an ZDXL355 accelerometer. The proposed approach is useful for practical applications of accelerometers. Overall, the manuscript quality is good, with detailed background introduction, calibration method description, experiments, and data analysis. I would suggest considering it for publication after addressing the following comments:

1.     The authors should rewrite the abstract to clearly summarize this work and highlight the novelty of this work. Please clearly describe the aim of this work, the proposed method for calibration and temperature compensation, the novelty of the proposed approach, and the advantages of the proposed method over other calibration & compensation approaches.

2.     Page 1, L33: The inaccuracy in the accelerometers will result in the inaccurate position measurement other than the angle estimation.  An inertial navigation system uses motion sensors (accelerometers) to measure acceleration, while uses rotation sensors (gyroscopes) to measure the angular velocity. The inaccuracy in the gyroscope will result in the inaccuracy in the tilt angle measurements.  

3.     Page 2, L 83, L 85: an accelerometer doesn’t measure force. Instead, it measures acceleration (a). If the force presents without creating acceleration to the suspended mass (m), an ideal accelerometer is expected to show no response. The presentence of the acceleration will create force on the mass (F=ma), which could lead to the mechanical deformation, and further variations in the electrical signals.

4.     Page 4, L 126, L 129, please change the labeling (1, 2) to avoid repetition with the section number, 1, 2, 3…

5.     The font size of labeling in figures 4-6, 9, 10 is very small, which is hard to read.

Reviewer 2 Report

Comments

In this paper, a method for calibration and thermal compensation of triaxial MEMS accelerometers is proposed. The title of the article is "Thermal Calibration", but a large part of the paper is devoted to the calibration of triaxial MEMS accelerometer frame errors, so the title of the article is inappropriate. In addition, it is not clear from the authors' description that there is a correlation between the frame and thermal errors, and the entire article needs to be restructured to improve the overall article. It is recommended that the authors carefully revise the paper. The following are specific comments.

1. The frame error modeling of MEMS accelerometers is introduced in a large part of the introduction, while there are only two papers on thermal error modeling, which is a small amount of literature and a limited introduction to the current state of research. It is suggested that the authors find more literature for a detailed review and carefully revise the introduction section.

2. The authors only describe two algorithms in Section 3, but how exactly they are modeled is not explained. In addition, the authors have directly chosen these two methods in the text without comparing them with other methods. It is suggested that the authors add algorithm flowcharts to describe how the two compensation algorithms compensate for structural and thermal errors.

3. From Figure 9(a), it can be seen that the thermal error of the accelerometer is linearly related to the temperature change. There are many algorithms for linear fitting, isn't it possible to achieve the selection of the best method for thermal error modeling by comparing the residuals of various fitting algorithms?

Round 2

Reviewer 2 Report

None.